# Acrylamide Hydrogel-Modified Silicon Nanowire Field-Effect Transistors for pH Sensing

**DOI:** 10.3390/nano12122070

**Published:** 2022-06-16

**Authors:** Gangrong Li, Qianhui Wei, Shuhua Wei, Jing Zhang, Qingxi Jin, Guozhi Wang, Jiawei Hu, Yan Zhu, Yun Kong, Qingzhu Zhang, Hongbin Zhao, Feng Wei, Hailing Tu

**Affiliations:** 1State Key Laboratory of Advanced Materials for Smart Sensing, GRINM Group Co., Ltd., Beijing 100088, China; gangrongli@foxmail.com (G.L.); jinqingxi@foxmail.com (Q.J.); wangguozhi0809@126.com (G.W.); zhuyancxy@foxmail.com (Y.Z.); yunkong0503@163.com (Y.K.); zhaohongbin@grinm.com (H.Z.); tuhl@grinm.com (H.T.); 2GRIMAT Engineering Institute Co., Ltd., Beijing 101407, China; 3General Research Institute for Nonferrous Metals, Beijing 100088, China; 4School of Information Science and Technology, North China University of Technology, Beijing 100144, China; weishuhua@ncut.edu.cn (S.W.); zhangj@ncut.edu.cn (J.Z.); hujiawei@ime.ac.cn (J.H.); 5Advanced Integrated Circuits R&D Center, Institute of Microelectronic of the Chinese Academy of Sciences, Beijing 100029, China; zhangqingzhu@ime.ac.cn

**Keywords:** acrylamide hydrogel, silicon nanowire, FET sensor, pH sensitive

## Abstract

In this study, we report a pH-responsive hydrogel-modified silicon nanowire field-effect transistor for pH sensing, whose modification is operated by spin coating, and whose performance is characterized by the electrical curve of field-effect transistors. The results show that the hydrogel sensor can measure buffer pH in a repeatable and stable manner in the pH range of 3–13, with a high pH sensitivity of 100 mV/pH. It is considered that the swelling of hydrogel occurring in an aqueous solution varies the dielectric properties of acrylamide hydrogels, causing the abrupt increase in the source-drain current. It is believed that the design of the sensor can provide a promising direction for future biosensing applications utilizing the excellent biocompatibility of hydrogels.

## 1. Introduction

pH detection is critical for a variety of chemical and biological applications, including enzyme catalysis [1], tumor monitoring [2], water quality detection, blood testing, etc. [3,4]. A device with high sensitivity, fast response, miniaturization, and portability is quite essential for pH monitoring in various environments and conditions [5]. Currently, pH sensors based on electrochemical [6], physical [7], and optical [8] mechanisms have been developed, among which the potentiometric method [9] based on electrochemical principles is widely used for pH sensing. However, the potentiometric pH sensors still have many disadvantages, which comprise difficulties in reference electrode miniaturization and potential instability during long-term operation.

Sensors based on the field-effect transistor (FET) technique have attracted much attention because of their excellent scalable and stable properties. Compared with traditional methods, silicon nanowire (SiNW) FET sensors have the advantages of small size, low cost, fast response, high sensitivity, label-free operation, and integration capability [10]. The pH detection principle of the FET is that the oxide surface groups can be charged due to the (de) protonation of the terminal OH groups at the interface in contact with the electrolyte. It can be described by the site-binding model [11], triggering the redistribution of the surface charge or potential of the silicon nanowire, ultimately leading to significant changes in drain current and voltage in changes of H^+^ concentration [12,13,14]. However, there is an equilibrium of the (de)protonation reaction of oxide OH groups at the local interface, and the results of the reaction directly affect the interface charge density [15]. The model can become quite complicated when surface-modified probe molecules are used for biosensing because of possible interference of protonation of OH groups, which may counteract the charge of the target molecule [16,17]. Especially, the complex environment of liquid testing for SiNW biosensors in the future may lead to the destruction of the insulating layer of SiNW sensors and an increase in leakage current.

Polymer hydrogel, with excellent biocompatibility and environmental tolerance has the ability to respond to stimuli from the surrounding environment, and immobilize chemicals and biomolecules through physical encapsulation or chemical bonding [1,18,19]. We propose that its film exhibits much promise to overcome these problems above and provides a new solution for a new generation of FET biosensors [20]. However, the problem with the current hydrogel-based FET is that the hydrogel still stays in the macroscopic size range and cannot match the size of the FET [21,22,23], leaving many questions to be answered.

Herein, we propose an innovative approach consisting of UV polymerization in situ to form the pH-responsive acrylamide hydrogel, modifying it on the gate of a SiNW FET. The synthetic method of acrylamide hydrogel is particularly interesting because it separates the polymerization of monomers and the cross-links between polymer chains for analysis [24,25]. Moreover, through the step transitions of dielectric properties of a nanometric layer interface, the electrical characteristics of pH response can be dynamically and in situ measured. Currently, few reports use the hydrogel polymer as a functional modification layer for SiNW FETs, which opens the door to interesting applications.

## 2. Materials and Methods

The SiNW FET sensors were fabricated based on the advanced 200 mm CMOS platform at the Institute of Microelectronics of the Chinese Academy of Sciences. Acrylamide (AAm), 3-(Trimethoxy silyl) propyl methacrylate (TMSPMA), 2-Hydroxy-4’-(2-hydroxyethoxy)-2-methylpropiophenone were purchased from Sigma-Aldrich. Sodium hydrogen phosphate (Na_2_HPO_4_, 99%), sodium dihydngen phoshate anhydrous (NaH_2_PO_4_, 99%), analytical-grade ethanol (C_2_H_5_OH, 99.5%), environmental-grade acetic acid (C_2_H_4_O_2_, 99%) were purchased from Alfa Aesar. hydrochloric acid (HCl, 30%), Sodium hydroxide (NaOH, 99%) were purchased from Innochem. Dialysis Membranes (2000) were purchased from Shanghai Yuanye Biotechnology Co., LTD, Shanghai, China.

SiNW FET Device Fabrication. Fabrication flow of SiNW FET is shown in Appendix A. SOI substrates were fabricated by depositing 145 nm SiO_2_ and 40 nm polysilicon on 200 mm silicon surface SiO_2_. Next, α-Si and SiNx thin films were deposited, and a rectangular pattern was formed through the steps of gluing, pre-baking, exposure, and development with the 4μm of pattern array spacing. The SiNx and α-Si were etched through a dry etching process to form side wall, which was approximately 90°. Then, the top SiNx hard mask (HMs) was subsequently removed in a hot H_3_PO_4_ solution at 1400 °C and SiNx film was deposited using a plasma-enhanced chemical vapor deposition (PECVD) process, followed by the corresponding silicon nitride reactive ion etching (RIE) to form two SiNx spacers on both sides of α-Si. Meanwhile, removed α-Si with tetramethyl hydroxid, and a photoresist of the corresponding shape of the pad was evenly spread on both ends to form a SiNW pattern with a support structure on the basis of the SiNx hard mask. Next, the SiO_2_ and Si at the bottom were etched sequentially by using the nanosized SiNx and the photoresist of the corresponding shape of the pad as a mask, and the photoresist was then removed by oxygen plasma bombardment, and the SiNx mask and top HMs were removed by hot H_3_PO_4_ and dilute hydrofluoric acid (DHF) solution to obtain the uniformly supported SiNW arrays. Last, using nickel-platinum alloy to form metal silicide in the source and drain regions, then implanting a large dose of low-energy boron (arsenic) ions, followed by high-temperature annealing (RTA) to activate the implanted impurity ions, then using photolithography, etching processes, oxygen plasma bombardment to remove photoresist to prepare source-drain electrodes. The fabrication method was the same as the previous method of the research group [26].

Synthesis and fixation of hydrogels. First, Acrylamide, 3-mercaptopropyl-trimethylsilane, glacial acetic acid solution, 3-methacrylate trimethoxysilyl propyl ester, and 2-hydroxy-4’-(2-hydroxyethoxy)-2-methylpropiophenone were mixed. Next, filled the solution into a plastic syringe and irradiated it under a UV lamp with a power of 6 W for 60 min, maintaining a distance of 5 cm between the sample and the bulb. Then, the photopolymerized transparent solution regenerated cellulose dialysis bag was placed in a beaker containing 1 L of deionized water and allowed to stand for 24 h. Meanwhile, the surface of the device was treated with a plasma degumming machine (Branson IPC 300, Branson, St. Louis, MO, USA), covered with hydroxyl groups on the surface, and a spin coater (KW-4A, Beijing, China) was used to homogenize under suitable process conditions of 500 rmp/min, 18 s, and 3500 rmp/min, 60 s. Finally, the spin-coated silicon wafers were placed in a container with saturated humidity and stored in an oven at 65 °C for 24 h to complete the fixation.

Characterization of hydrogel. Gate functionalization was monitored using infrared spectrometer (FTIR, Thermo Scientific Nicolet iS20 Nicolet iS50, Thermo Scientific, Waltham, MA, USA) and scanning electron microscope (SEM, Hitachi S-5500, Hitachi, Tokyo, Japan), and the white porous solid after freeze-drying of the hydrogel solution was shown in Appendix A.

Measurement and analysis of SiNW FET devices. Electrical measurements of the SiNW FET sensing device were performed using the Keithley 4200A-SCS semiconductor parameter analyzer. The SiNW FET is characterized when the gate sensitive area is immersed for a few seconds in an electrolytic solution, then an electrical curve is acquired. The same measurement is repeated on the same devices for six different pH values, namely, 3, 5, 7, 9, 11, and 13, and the configuration process of pH solutions is shown in supporting information. The gate is thoroughly rinsed with deionized water after each measurement and the ionic strength of the different pH solutions is always kept constant. To evaluate the performance of the device, the relationship between the drain current (I_D_) and the gate voltage (V_G_), and the relationship between the drain current (I_D_) and the drain voltage (V_D_) was explained. To measure the real-time electrical response of the SiNW FET sensor, we applied constant V_D_ and V_G_ to the device during the measurement and recorded a sync I_D_ every less than 1 s to avoid thermal drift of the FET device. The recorded current can be used to observe the response of the SiNW FET sensor in the buffer.

Capacitance Measurement. The capacitance measurement was performed using electrochemical Impedance Spectroscopy (EIS). Experiments were performed in a three-electrode cell using an electrochemical workstation (Bio-Logic VMP3e, Seyssinet-Pariset, France) with a hydrogel-modified platinum disk electrode as the working electrode, a commercial saturated calomel reference electrode, and a large platinum grid (5 cm^2^) as auxiliary electrode. The impedance spectrum was analyzed using a nonlinear fit (Zview, EIS Spectrum Analyzer free software, Southern Pines, NC, USA) to the electrical behavior of the equivalent circuit shown in Appendix A.

## 3. Results and Discussion

Figure 1a shows the cross-sectional TEM image and the electron scattering spectrum (EDS) elemental mappings of the fabricated SiNW. According to the TEM and SEM images (Appendix A), the thicknesses of the HfO_2_/SiO_2_ layers are about 9.96 nm and 2.49 nm, respectively. And the prepared SiNW is about 25.86 nm wide and 36.39 nm high, and the SiNW arrays are uniformly arranged. In addition, the EDS analysis of Si, Hf, and O elements shows that the interface of the HfO_2_ and SiO_2_ gate dielectric insulating layers is clear and flat. The gate dielectric layer completely wraps the silicon wire to reduce leakage current from the liquid to the device and provide a robust liquid gate environment, which is illustrated in Figure 1b.

As depicted in Figure 1c, the functionalization strategy consists of substrate preparation, coating, and curing. FTIR shows that in the process of solution configuration, the addition of TMSPMA will react with AAm and other TMSPMA, according to the red shift of the absorption peak of the Si-O bond in the infrared spectrum of nanogel. Figure 2 shows that the lyophilized gel has a large number of voids after swelling, and this high-porosity structure is conducive to the transport and diffusion of molecules and ions. Moreover, the SEM mapping diagram shows a similar element distribution state and all elements distributed uniformly on the sample, indicating that the silane coupling agent successfully reacts with acrylamide so that the gel formed a cross-linked three-dimensional porous structure.

To examine the FET before and after hydrogel modification, we tested the basic electrical properties of the hydrogel functionalized SiNW FET and the SiNW FET. The SiNW FET used ALD to deposit 10 nm of HfO_2_ prior to testing. Figure 3 shows the I_DS_–V_GS_ and I_DS_-V_DS_ curves by bias gate voltages of the hydrogel functionalized sensor, which exhibits a typical p-type field-effect behavior as well as the SiNW FET (Appendix A). For the electrical characteristic curve of V_DS_ = 2 V, the corresponding extracted values of switching ratio and subthreshold swing (SS) are estimated to be 1.03 × 10^4^ and 1.93 V/dec, respectively.

To evaluate the pH characteristics, the hydrogel functionalized SiNW FET was detected by immersing the gate-sensitive region in a buffer pH solution for a few seconds (Appendix A). A part of the SiNW FET before modification was observed to leak electricity when adding a liquid gate. The relationship between I_GS_ and transfer characteristics is shown in Appendix A. The leakage current between source and drain of the SiNW FET is much larger than that of the hydrogel-modified device. The typical sensing characteristics of the silicon nanowire sensors under different pH gate environments are shown in Figure 4a,c. The drain current increases when the pH increases from 3 to 13, leading to certain shifts in the threshold voltage (V_th_). The threshold voltages are estimated by the constant current method for the SiNW FET and the hydrogel-modified SiNW FET, respectively [26,27]. The voltage corresponding to the current of 3.90625 × 10^−9^ is the threshold voltage. The extracted voltages are shown in Figure 4b,d, with a linear relationship with pH. The formula for calculating sensitivity is as follows: (1)$$S=\frac{V_{\mathrm{th}1}-V_{\mathrm{th}2}}{{\mathrm{pH}}_{1}-\;{\mathrm{pH}}_{2}}\;$$

S is the sensitivity of the device. For the SiNW FET, the average sensitivity is 68 mV/pH; meanwhile, the average sensitivity of the polymer-hydrogel-coated device is 100 mV/pH, which is more sensitive than the reported polyvinyl alcohol/polyacrylic acid (PVA/PAA) hydrogel nanofiber potentiometric sensor with a pH sensitivity of 74 mV/pH [28]. 

The V_th_ depends on the gate capacitance according to literature reports [29,30]. Instead of HfO_2_, the dielectric layer of the hydrogel-modified SiNW FET is an acrylamide ion hydrogel film. Capacitance versus frequency of ionic hydrogels can be evaluated by EIS. Under the hypothesis of a totally blocking electrode without faradaic reactions, the imaginary part Z_im_ of the total impedance is almost purely capacitive, so capacitance can be expressed according to the following formula: C = −1/(ωZ_im_) (Appendix A) [31]. Capacitance values at very low frequencies are consistent with the dc values extracted from the overall impedance spectra fitting [19,32].

Figure 5 shows the plot of capacitance as a function of buffer pH. It can be obviously noted that the capacitance of hydrogel-functionalized electrodes shows a close to linear increase when the pH varies from acidic to basic values. When the acrylamide hydrogel swelled in equilibrium in the pH range of 0–13, the ionization degree of the carboxylic acid group gradually increased with the increase in buffer pH. Due to the large increase in the number of fixed charges, the ability to attract mobile ions increases in the polymer network instead of the gate dielectric surface. It will result in a large ion concentration difference between the inside and outside of the hydrogel, which means that the osmotic pressure of the hydrogel-modified SiNW FET will greatly exceed that of the external solution. The swelling ratio of the hydrogel changes the capacitance of the gate as the pH value increases, resulting in a large shift in the threshold voltage for operating the SiNW FET [33,34]. Therefore, the pH sensitivity of the hydrogel-modified SiNW FET significantly increases according to Equation (1).

The real-time response of I_DS_ is shown in Figure 6, in which the drain current markedly increases when dropping a liquid solution on the gate. When the pH value of the buffer changes from acidic to basic, the time change rate curve changes regularly. The results are consistent with the transfer curve of the p-type silicon nanowire sensor increasing. The rate of change of different buffer pH curves is extracted from the time curve, and its distribution is shown in Figure 6b,d. For the SiNW FET, the rate of change ranges from 168 to 396, which is linear with pH correlation. While the rate of change ranges from 179 to 14,520 for hydrogel-modified SiNW FET devices, it is exponentially related to the value of buffer pH. Therefore, when the ion concentration of the liquid to be tested is constant, the change in pH for the hydrogel-modified SiNW FET sensors is more sensitive.

Figure 7 depicts the transfer and real-time response curves after placing the above devices quietly in a clean room for four weeks. The hydrogel-modified SiNW FET still shows a certain regularity with a pH sensitivity of 80.03 mV/pH (R^2^ > 0.99), with an excellent linear correlation, while the SiNW FET is 30.68 mV/pH. Compared with Figure 4, the pH sensitivity of the hydrogel-modified SiNW FET is reduced by about 20%, and the SiNW FET is reduced by nearly 50%, which means that the polymer film of tens of nanometers on the SiNW FET protects the gate well. Meanwhile, Figure 7c,d shows the regularly and repeatedly electrical signal response of the hydrogel-modified SiNW FET in different buffer pH solutions, and the current rate of change ranges from 58 to 12,339 with an excellent exponential correlation (R^2^ > 0.99).

Table 1 lists a pH sensitivity comparison of the relevant reported FET sensors [13,19,35,36,37]. The results show that the hydrogel-functionalized SiNW FET sensors have a satisfactory performance to pH sensors prepared by other methods and can be applied in future pH sensing instant applications.

## 4. Conclusions

In this work, hydrogel-functionalized SiNW FETs have been successfully fabricated, which show excellent pH sensing performance with an average pH sensitivity of 100 mV/pH. And the rate of change for hydrogel-modified SiNW FET devices is exponentially related to the value of pH. A mechanism is proposed to explain why SiNW FET with hydrogel as the dielectric layer is more sensitive to pH detection. Swelling of hydrogels occurring at the gate surface modifies dielectric properties of hydrogels, causing in turn measurable variations of the drain current. It is considered that the hydrogel-modified SiNW FET delays device aging and possesses excellent stability. Our study, therefore, provides a new approach to the development of a novel class of versatile, highly sensitive, and real-time response biosensors for detection in aqueous media.

## Figures and Tables

**Figure 1 nanomaterials-12-02070-f001:**
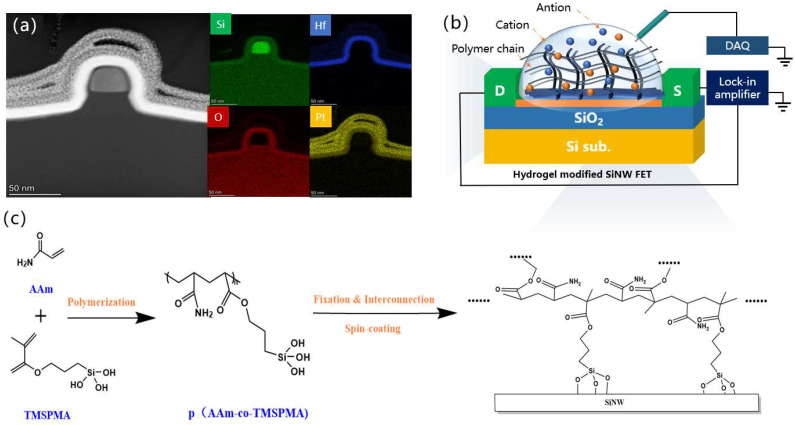
(**a**) TEM image of Si NW channel of the device, the right: EDS elemental mappings of O, Si, Hf and Pt, respectively. (**b**) Schematic test of hydrogel-modified SiNW FET surface. (**c**) Strategy for the modification of the gate, including polymerizing acrylamide to form p(AAm-co-TMSPMA), spin-coating and curing of hydrogels onto FET surfaces.

**Figure 2 nanomaterials-12-02070-f002:**
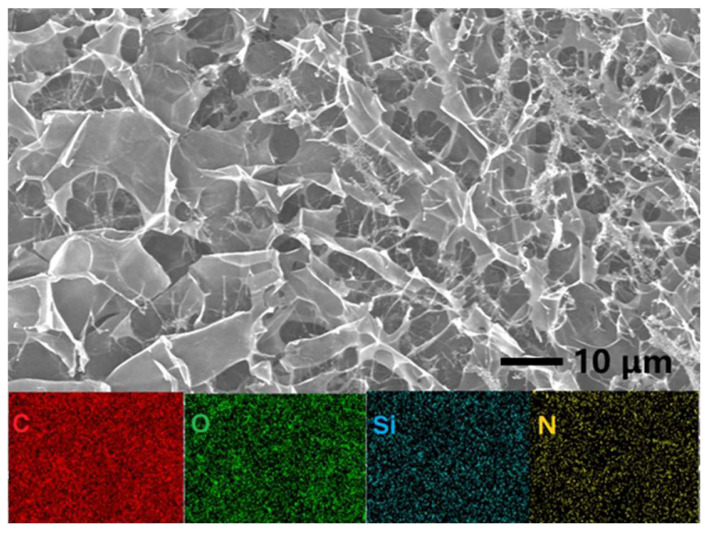
SEM image of freeze-drying of the hydrogel solution, and EDS elemental mappings of C, O, Si, and N, respectively.

**Figure 3 nanomaterials-12-02070-f003:**
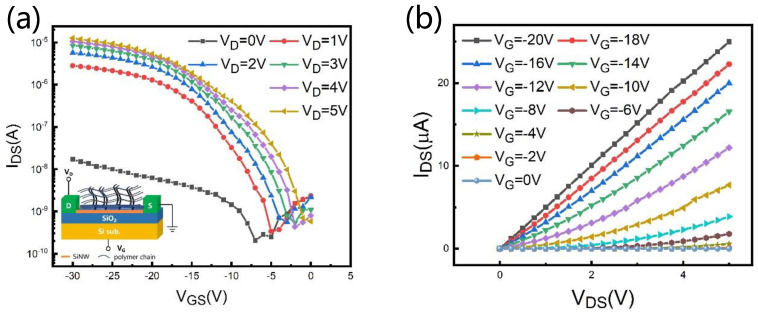
Electrical characteristic curve of SiNW FET after polymer-hydrogel coating, (**a**) Transfer characteristic curve (I_DS_–V_GS_), the inset is a schematic diagram of device detection. (**b**) Output characteristic curve (I_DS_–V_DS_).

**Figure 4 nanomaterials-12-02070-f004:**
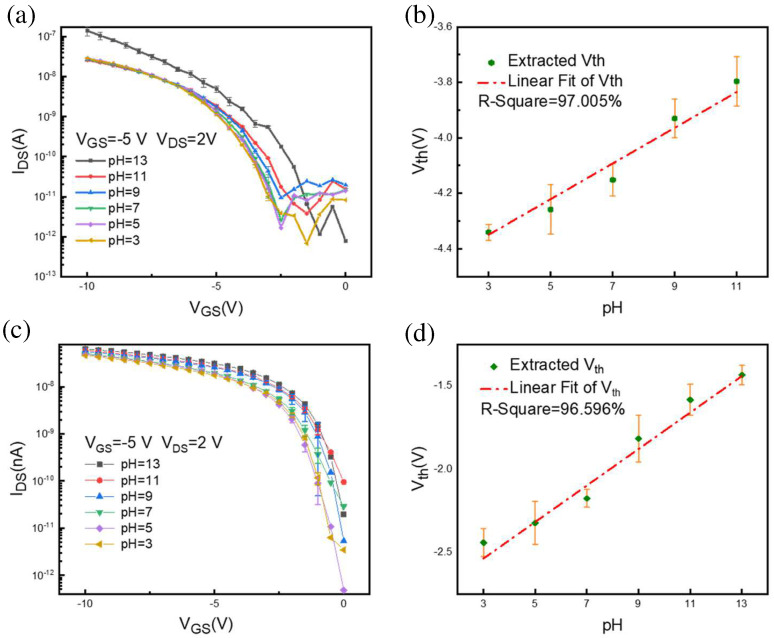
(**a**) Transfer characteristic curve (I_DS_–V_GS_) of SiNW FET at different pH liquid gate environments (3–13), (**b**) extracted threshold voltage under different pH time-response curves of SiNW FETs. (**c**) Transfer characteristic curve (I_DS_–V_GS_) of hydrogel-functionalized SiNW FET at different pH liquid gate environments (3–13), (**d**) dot plot of extracted threshold voltage under different pH time-response curves of hydrogel-functionalized SiNW FETs.

**Figure 5 nanomaterials-12-02070-f005:**
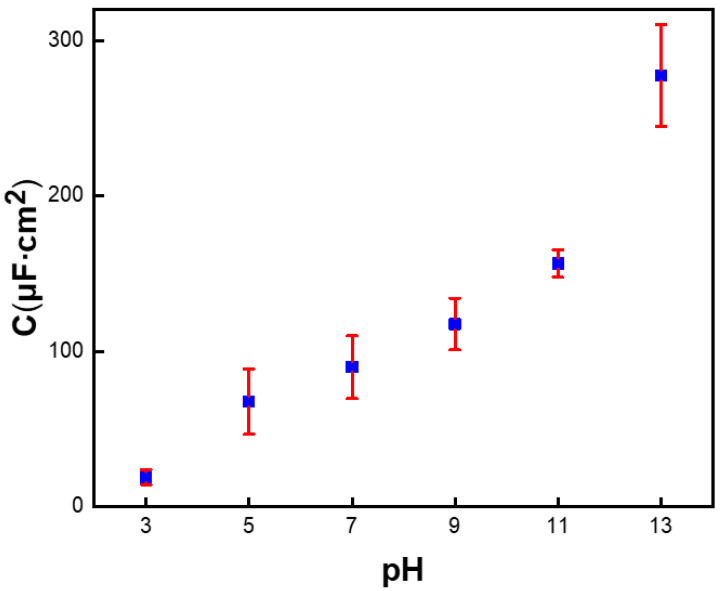
Capacitance as a function of pH for hydrogel-functionalized electrodes.

**Figure 6 nanomaterials-12-02070-f006:**
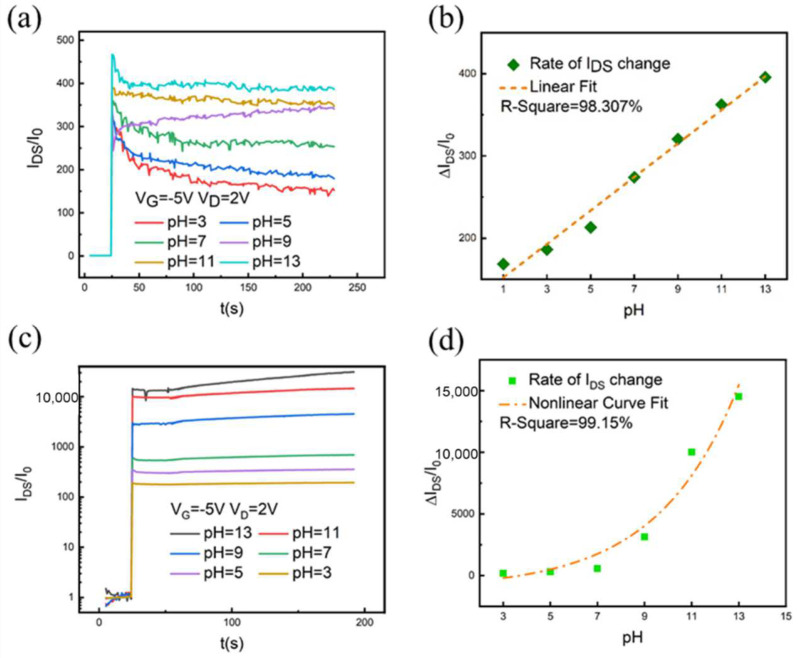
(**a**) Real-time response curve of I_DS_ as a function of time for SiNW FET in different pH solutions, and (**b**) Current rate of change under different pH time-response curves of SiNW FETs. (**c**) Real-time response curve of I_DS_ as a function of time for hydrogel-functionalized SiNW FET in different pH solutions, and (**d**) Current rate of change under different pH time-response curves of hydrogel-functionalized SiNW FETs.

**Figure 7 nanomaterials-12-02070-f007:**
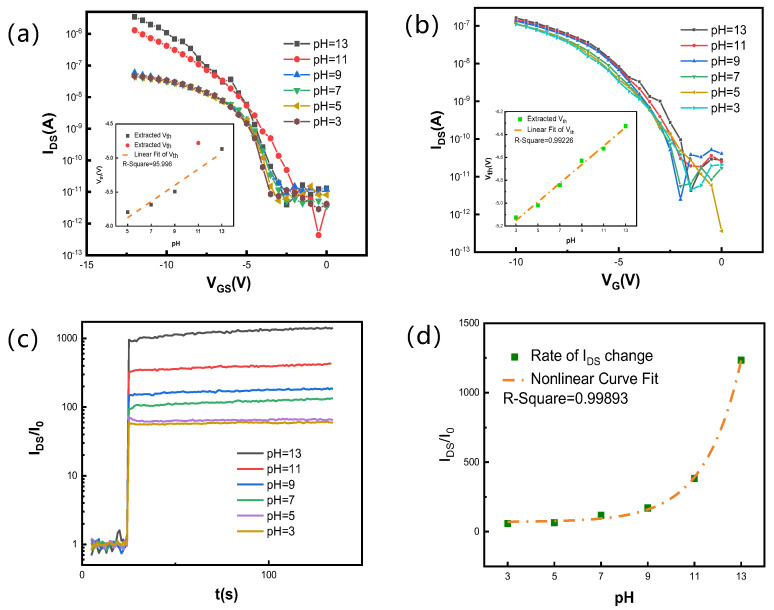
(**a**,**b**) Transfer characteristic curve (I_DS_-V_G_) at different pH liquid gate environments (3–13) of SiNW FET and hydrogel-functionalized SiNW FET after four weeks, respectively, and the insets are extracted threshold voltage under different pH time-response curves. (**c**) Real-time response curve of I_DS_ as a function of time for hydrogel-functionalized SiNW FET in different pH solutions after four weeks, and (**d**) current rate of change under different pH time-response curves of hydrogel-functionalized SiNW FET.

**Table 1 nanomaterials-12-02070-t001:** Comparison of FET pH Sensors.

pH Sensitivity	Nanowire Materials	Ref.	Published Journal Information
42.2 nA/pH	Si nanobelt	Chi-Chang et al. [35]	Sensors, 2021
178 mV/pH	Si nanowire	Siqi et al. [13]	Nanomaterials, 2020
84.8 mV/pH	EGFET	Ghoneim et al. [36]	Small Methods, 2018
——	Hydrogel-Gated OFET	Laure et al. [19]	Langmuir, 2018
−57.66 mV/pH	ISFET	Lee et al. [37]	Sensors and Actuators, B Chemical, 2015
100 mV/pH	Hydrogel-Gated Si nanowire	This work	——

## Data Availability

Not applicable.

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
