# Peer review of "Acrylamide Hydrogel-Modified Silicon Nanowire Field-Effect Transistors for pH Sensing"

_nanomaterials, 2022, doi:10.3390/nano12122070_

Round 1

Reviewer 1 Report

In this work, authors report a pH-responsive hydrogel modified silicon nanowire field-effect transistor for pH sensing. The work presents different experimental implementations of SiNWs acting as highly sensitive pH sensors. However, there are several aspects of this paper that should be properly modified to be accepted.

1.- Wording must be improved, there are several incorrect sentences that create confusion. As an example, we can mention: “…Currently few reports using the hydrogel polymer as a functional modification layer for SiNW FETs, which opens…”

2.- Authors mention in the Results section that the HfO2/SiO2 layers have some thickness, and both materials are depicted in Figure 1. However, in the SiNW FET Device Fabrication Section there is no reference to HfO2 or the process employed to create that insulator layer. Apparently you have used a process without HfO2 but here this material plays an important role. This aspect is hard to understand and should be clearly explained.

3.- Sensitivity is defined in Eq. (1) and then it is claimed that “…the average sensitivity is 68mV/pH…” . This value is obtained for the standard device, without polymer hydrogel coating the surface. However, this value is surpassing the Nernst limit, established in 59.2 mV/pH as it is well known from classical references [1]. An explanation for this apparently surprising result must be provided, clarifying the differences with previous theories.

4.- In this sense, the high sensitivity provided by the hydrogel is due to “…The swelling ratio of the hydrogel changes the capacitance or dielectric constant of the gate as the pH value increased, resulting in a large shift in the threshold voltage for operating the SiNW FET[34,35]”. This explanation is similar to those already provided in previous research works but in this case, they are not supported by any experimental evidence, as it could be, for example, some type of measurement of the physical properties of the hydrogel, such as its dielectric constant. A further elaboration in this direction should be addressed, instead of just repeating previous hypothesis.

[1] Advances in Colloid and Interface Science, 69 (1996) pp. 31-62

A general model to describe the electrostatic at electrolyte oxide interfaces

R.E.G. van Hal, J.C.T. Eijkel, P. Bergveld

Author Response

Response to Reviewer 1 Comments

Point 1: Wording must be improved, there are several incorrect sentences that create confusion. As an example, we can mention: “…Currently few reports using the hydrogel polymer as a functional modification layer for SiNW FETs, which opens…”

Response 1: Thanks for pointing it out. We had proof-read the manuscript carefully and corrected the formatting errors in the manuscript. The location of the change is as follows.

Corresponding change in manuscript: Yes
Location of Change:
Section: Introduction
Page 2 from line 33-34.

Location of Change:
Section: Materials and Methods
Page 3 from line 7-8.
Page 3 from line 9.

Location of Change:
Section: Results and discussion
Page 5 from line 9.
Page 6 from line 9-10.
Page 9 from line 3.

Point 2: Authors mention in the Results section that the HfO2/SiO2 layers have some thickness, and both materials are depicted in Figure 1. However, in the SiNW FET Device Fabrication Section there is no reference to HfO2 or the process employed to create that insulator layer. Apparently you have used a process without HfO2 but here this material plays an important role. This aspect is hard to understand and should be clearly explained.

Response 2: We added a description of the steps to deposit HfO2 prior to testing and characterizing hydrogel unmodified SiNW-FET.

The HfO2 oxide layer is mainly used as insulation, so that the device can be stable to complete the test after the addition of the liquid grid. We typically use ALD to deposit 10 nm of HfO2 prior to sensing testing with SiNW-FETs. We notice that polycrylamide is almost non-conductive and acts as insulation[1]. We can see from Figure S5 that when testing the performance of the liquid gate, the value of the IGS is much smaller than the IDS. Originally, we wanted to show the TEM image of the hydrogel-modified SiNW-FET, but the softening temperature of polyacrylamide is 210 °C[2], so the original structure could not be maintained after the sample preparation by focused ion beam(FIB)
sectioning.

TEM cross-sectional view of the hydrogel-modified SiNW-FET.
[1] Fuchang Sun, Yuchen Pan, Yunfei Zhang, Hui Liu, Feipeng Du. Preparation and properties of PSS/ poly (acrylamide-methacrylic acid) conductive hydrogel [J]. Journal of Composite Materials,202,39(03):1131-1140. (Chinese)
[2] He Tielin. Handbook of Water Treatment Chemicals. Beijing: Chemical Industry Press, 2000.05:81(Chinese)

Corresponding change in manuscript: Yes
Location of Change:
Section: Results and discussion
Page 6 from line 2.

Point 3: Sensitivity is defined in Eq. (1) and then it is claimed that “…the average
sensitivity is 68mV/pH…” . This value is obtained for the standard device, without polymer hydrogel coating the surface. However, this value is surpassing the Nernst limit, established in 59.2 mV/pH as it is well known from classical references [1]. An explanation for this apparently surprising result must be provided, clarifying the differences with previous theories.

Response 3: Thanks for your comments. First of all, in the Nernst limit theory[1], the dielectric constant of HfO2 is between Al2O3 and Ta2O5. Under this theory, α is between 0.9 and 0.95, means that the sensitivity with HfO2 is close to 58mV/pH.

“…the observed sensitivity below 59.2 mV/pH, it was initially concluded that surface reactions between the gate insulator and the electrolyte solution should determine the primary response mechanism…”[1]. The Nernst limit model, which mainly considers the ion transfer process during the (de)protonation reaction of the surface, resulting in the change of the surface charge. Recently, we also noted that Zhong Lin Wang's team found that electron transfer exists in the process of solid-liquid contact, which confirmed that the electrification of solid-liquid contact was the common result of electron transfer and ion transfer from both microscopic[2] and macroscopic[3] perspectives.

In a word, the Nernst limit theory before is no longer in line with the actual situation.

[1] R.E.G. van Hal, J.C.T. Eijkel, P. Bergveld. A general model to describe the electrostatic at electrolyte oxide interfaces. Advances in Colloid and Interface Science, 69 (1996) pp. 31-62.
[2] Shiquan Lin, Liang Xu, Aurelia Chi Wang, Zhong Lin Wang*. Quantifying electrontransfer in liquid-solid contact electrification and the formation of electric double-layer Nat. Commun., 2020, 11, 399, DOI: 10.1038/s41467-019-14278-9.
[3] Fei Zhan, Aurelia C. Wang, Liang Xu, Shiquan Lin, Jiajia Shao, Xiangyu Chen*, and Zhong Lin Wang*.Electron Transfer as a Liquid Droplet Contacting a Polymer Surface[J]. ACS Nano, 2020, 14(12). DOI: 10.1021/acsnano.0c08332.

Corresponding change in manuscript: No

Point 4: In this sense, the high sensitivity provided by the hydrogel is due to “…The swelling ratio of the hydrogel changes the capacitance or dielectric constant of the gate as the pH value increased, resulting in a large shift in the threshold voltage for operating the SiNW FET[34,35]”. This explanation is similar to those already provided in previous research works but in this case, they are not supported by any experimental evidence, as it could be, for example, some type of measurement of the physical properties of the hydrogel, such as its dielectric constant. A further elaboration in this direction should be addressed, instead of just repeating previous hypothesis.

Response 4: our comments. Hydrogel is a kind of soft substance, which is composed of a hydrophilic polymer network. Hydrogel can absorb a lot of water and expand to twice its own volume to more than a thousand times its volume. In the network structure, water molecules move relative to each other and behave as a liquid; the polymer network deforms but remains cross-linked and behaves as a solid. This solid-liquid property makes hydrogel a unique class of materials. The irregular shape of the hydrogel makes it difficult to calculate its dielectric constant from its capacitance. We delete the word "dielectric constant" in the manuscript to make the article description more rigorous.

Corresponding change in manuscript: Yes

Location of Change:

Section: Results and discussion

Page 8 from line 3-6.

Reviewer 2 Report

Authors have fabricated a hydrogel-functionalized SiNW FET for pH sensing. While the study is of broader interest to the journal readers, it should be improved in few aspects:

1) With respect to the SEM image in figure 2 and the porosity of the polyacrylamide hydrogel: could the authors clarify the chemical entities  (and real samples) vs pH that can be sensed? The results should be summarized in a Table.

Can the average sizes of the pores be modified and what are the possible consequences? Is it possible to follow a titration curve?

2) Authors have provided a schema in Figure 2b. Could a picture of the real pH meter be provided?

3) With respect to the Pt and Hf elemental presence (SEM figures): it is not clear the function of these elements and how these were introduced in the layers described in figure 1a.

Author Response

Response to Reviewer 2 Comments

Point 1: With respect to the SEM image in figure 2 and the porosity of the polyacrylamide hydrogel: could the authors clarify the chemical entities (and real samples) vs pH that can be sensed? The results should be summarized in a Table.

Can the average sizes of the pores be modified and what are the possible consequences? Is it possible to follow a titration curve?

Response 1: We added freeze-dried SEM images of hydrogels swollen at different pH conditions. We found that the hydrogel exhibited a point-like porous structure when the pH value is 3. While the hydrogel exhibited a lamellar porous structure when pH value is 7 or 13, greatly increasimg the swollen volume of the hydrogel and the size of the mesh size. It would increase the diffusion of solution ions. Figure S4 presents the actual sample of the white porous solid after lyophilization of the hydrogel solution. We could only see the surface topography and structure in two dimensions, so it was difficult to obtain an accurate porosity of the hydrogel. As for titration curves, we directly used a pH meter to titrate the solution. For example, the step to configure buffer that the pH value is 3 was to directly add HCl to NaH2PO4 solution, and using NaH2PO4 solution and Na2HPO4 solution configured buffer that the pH value is 7. The configuration method was different, so we did not measure the titration curve.

Corresponding change in manuscript: Yes

Location of Change:

Section: Supporting Informpation

Figure S4.

Point 2: Authors have provided a schema in Figure 2b. Could a picture of the real pH meter be provided?

Response 2: Thanks for your comments. We put the actual test environment in the supplementary information as figure S6.

Corresponding change in manuscript: Yes

Location of Change:

Section: Supplementary Information

Figure S6.

Point 3: With respect to the Pt and Hf elemental presence (SEM figures): it is not clear the function of these elements and how these were introduced in the layers described in figure 1a.

Response 3: Thanks for your comments. The Hf element is introduced as the gate insulating layer to guarantee that the device can be stable to complete the test after the addition of the liquid grid. And the Pt is introduced by platinum spraying because of poor surface conductivity when we go to take the TEM.

Corresponding change in manuscript: No

Reviewer 3 Report

 In this study, the authors report a hydrogel-modified silicon nanowire field-effect transistor for pH sensing. They show that the hydrogel sensor can measure buffer pH in a repeatable and stable manner in the pH range of 3-13, with a high pH sensitivity.

The study is interesting from both a fundamental and an application viewpoint.

The paper is clear and enough detailed. The study is well conducted and reports convincing results.

I support the publication of the paper after a minor revision.

Here are a few suggestions for the authors:

  1. SiNW FET Device Fabrication: Although quite detailed, I would suggest accompanying the description with a schematic drawing that illustrates the main fabrication steps.
  2. Figure 3 shows that the SiNW FET exhibits a typical p-type field effect behavior. Is there any explanation for it?
  3. “The threshold voltages are estimated (constant current method) for the SiNW FET and the hydrogel modified SiNW FET, respectively” Add the constant current value used to define Vth.
  4. “The Vth depends on the gate capacitance according to literature reports” Although references are given, for self-consistency, I suggest adding the equation that shows the dependence of Vth on the gate capacitance.

Author Response

Response to Reviewer 3 Comments

Point 1: SiNW FET Device Fabrication: Although quite detailed, I would suggest accompanying the description with a schematic drawing that illustrates the main fabrication steps.

Response 1: Thanks for your comments. We put the fabrication flow of SiNW-FET in the supplementary information as figure S1.

Corresponding change in manuscript: Yes

Location of Change:

Section:  Supplementary Information

Figure S1.

Location of Change:

Section: Materials and Methods

Page 3 from line 4-5.

Point 2: Figure 3 shows that the SiNW FET exhibits a typical p-type field effect behavior. Is there any explanation for it?

Response 2: Thanks for your comments. The source-drain current of the sensor increases exponentially with the negative back-gate voltage. After the threshold region is exceeded, the current increases slowly. This is because the majority carriers in P-type devices are holes, indicating that the device exhibits good characteristics of P-type devices

Corresponding change in manuscript: No

Point 3: The threshold voltages are estimated (constant current method) for the SiNW FET and the hydrogel modified SiNW FET, respectively” Add the constant current value used to define Vth.

Response 3: Thanks for your suggestion.We use the formula to calculate the current as 3.9062510-9A, and the voltage corresponding to this current is the threshold voltage. Please see the PDF for the formula

Corresponding change in manuscript: Yes

Location of Change:

Section:  Results and discussion

Page 6 from line 19-20.

Point 4: “The Vth depends on the gate capacitance according to literature reports” Although references are given, for self-consistency, I suggest adding the equation that shows the dependence of Vth on the gate capacitance.

Response 4: Thanks for your suggestion. For n-type FD (fully depleted) SOI MOSFETs, the relationship between Vth and gate capacitance is as follows[1]

Please see the PDF for the formula. The device to which the above formula applies was different from the device fabricated by us. We did not show this formula in the manuscript

[1] Zeng S R. Fundamentals of Semiconductor Device Physics. Beijing: Peking University Press, 2009

Corresponding change in manuscript: No

Round 2

Reviewer 2 Report

The authors have sent a detailed response to the issues raised by reviewer #2. However, I was unable to inspect the supplementary information where the new info is stored. The authors have elaborated and indicated where the info should be found.

Author Response

Response to Reviewer 2 Comments

Point 1: The authors have sent a detailed response to the issues raised by reviewer #2. However, I was unable to inspect the supplementary information where the new info is stored. The authors have elaborated and indicated where the info should be found.

Response: Thanks for pointing it out. We are very sorry that we forgot to mark the explanation for the swelling phenomenon of polyacrylamide hydrogels under different pH solutions. We have highlighted the revised information in Supplementary Materials.

Corresponding change in manuscript: Yes

Location of Change:

Section: Supporting Informpation

Figure S4 and page 3 from line 1-4

I have placed questions and answers that you may have questions about:

Point : With respect to the SEM image in figure 2 and the porosity of the polyacrylamide hydrogel: could the authors clarify the chemical entities (and real samples) vs pH that can be sensed? The results should be summarized in a Table.

Can the average sizes of the pores be modified and what are the possible consequences? Is it possible to follow a titration curve?

Response : We added freeze-dried SEM images of hydrogels swollen at different pH conditions. We found that the hydrogel exhibited a point-like porous structure when the pH value is 3. While the hydrogel exhibited a lamellar porous structure when pH value is 7 or 13, greatly increasimg the swollen volume of the hydrogel and the size of the mesh size. It would increase the diffusion of solution ions. Figure S4 presents the actual sample of the white porous solid after lyophilization of the hydrogel solution. We could only see the surface topography and structure in two dimensions, so it was difficult to obtain an accurate porosity of the hydrogel. As for titration curves, we directly used a pH meter to titrate the solution. For example, the step to configure buffer that the pH value is 3 was to directly add HCl to NaH2PO4 solution, and using NaH2PO4 solution and Na2HPO4 solution configured buffer that the pH value is 7. The configuration method was different, so we did not measure the titration curve.
